# Epidemiological Survey of the Main Tick-Borne Pathogens Infecting Dogs from the Republic of Moldova

**DOI:** 10.3390/pathogens11111267

**Published:** 2022-10-30

**Authors:** Gianluca D’Amico, Angela Monica Ionică, Adriana Györke, Mirabela Oana Dumitrache

**Affiliations:** 1Department of Parasitology and Parasitic Diseases, Faculty of Veterinary Medicine, University of Agricultural Sciences and Veterinary Medicine, 400372 Cluj-Napoca, Romania; 2Clinical Hospital of Infectious Diseases of Cluj-Napoca, 400348 Cluj-Napoca, Romania

**Keywords:** tick-borne diseases, tick-borne pathogens, dog, Republic of Moldova

## Abstract

Despite the significant burden of tick-borne diseases (TBDs), epidemiologic studies are missing, and TBD awareness is low in the Republic of Moldova. Our study is the first to assess the prevalence of the main tick-borne pathogens (TBPs) infecting dogs in this country and associated risk factors. In this cross-sectional, multi-centre study (June 2018–July 2019), blood samples were collected from dogs presenting in veterinary clinics (Chişinău: N = 30) and hosted in public dog shelters (Cahul: N = 42; Chişinău: N = 48). TBPs were assessed by molecular techniques and risk factors by the logistic regression model. *Hepatozoon canis* was the most prevalent TBP (15.8% [19/120]), followed by *Babesia canis* (11.7% [14/120]), *Anaplasma phagocytophilum* (5.8% [7/120]), and *Bartonella* spp. (0.8% [1/120]). Blood samples tested negative for *Borrelia* spp., *Rickettsia* spp., *Francisella tularensis*, *Anaplasma platys*, and *Ehrlichia canis*. Dogs originating from the veterinary clinics had a higher prevalence of *A. phagocytophilum* infection than those from the shelters (16.6% versus 2.2%, respectively, *p* = 0.0292; OR: 27.0 [95%CI: 1.4–521.9]). Dogs from Chișinău had a higher prevalence of *Hepatozoon canis* infection versus those from Cahul (19.2% versus 9.5%, respectively, *p* = 0.0295; OR: 3.9 [95%CI: 1.1–13.4]). We recommend routine use of acaricides and deworming of dogs to prevent or/and limit TBD spread. Further TBD surveillance studies are needed.

## 1. Introduction

Ticks, order Ixodida, are obligate hematophagous ectoparasites of mammals, birds, and reptiles, and some of them pose a great risk to both animal and public health [1]. Pathogens transmitted by ticks, such as bacteria, viruses, rickettsia, and protozoa, cause most of the vector-borne diseases reported in the temperate regions of Europe, North America, and Asia [2,3]. To date, growing evidence suggests that tick dissemination and tick-borne disease (TBD) transmission is expanding across European countries [4,5,6,7], becoming a significant issue due to individual and societal burdens (e.g., Lyme disease) [8], threats to animal health (both companion and livestock), and financial losses caused by debilitating or lethal TBDs, especially in developing countries (e.g., babesiosis) [2,9].

In the current context, disease surveillance, defined as the systematic collection, analysis, and dissemination of data on infections, plays a key role in planning and implementing suitable actions that can be taken to prevent or limit the spread of TBDs in a specific region [7].

Different studies have shown that *Ixodes ricinus* (Linnaeus, 1758) and *Dermacentor reticulatus* (Fabricius, 1794) are the most common ixodid ticks (Acari: Ixodidae) found in dogs across central and eastern European countries [10,11]. *Ixodes ricinus* is a suitable vector for zoonotic agents such as *Borrelia burgdorferi* sensu lato (*B. burgdorferi* s. l.), *Anaplasma phagocytophilum, Francisella tularensis*, tick-borne encephalitis virus, *Babesia canis,* and *Bartonella henselae,* while *D. reticulatus* may transmit *B. canis*, *Rickettsia* spp. and *A. phagocytophilum* [3,10,12]. Even if exposure to tick-borne pathogens (TBPs) does not always result in clinical disease in animals and humans, a greater awareness of TBDs in health professionals, veterinary doctors, and the general population is the first step for better management of patients and affected animals.

In the Republic of Moldova, TBPs were evaluated only in ticks collected from migratory birds and questing ticks. *A. phagocytophilum*, *Borrelia* genospecies, and spotted fever group *Rickettsia* were detected in *I. ricinus* ticks parasitizing birds [13,14,15] and those from the environment [16]. These findings raise further important questions: What are the infection rates in domestic animals? Are TBDs a risk for animal health? A significant amount of data provide confirmation that dogs may significantly contribute to the circulation of ticks and TBPs in the environment [17]. Despite the medical importance of some TBDs, epidemiological studies in dogs are lacking in this country. 

To address this gap in knowledge, our study aimed to assess the prevalence of the main TBPs infecting dogs from the Republic of Moldova and identify associated risk factors.

## 2. Materials and Methods

### 2.1. Study Design and Sampling

This was a cross-sectional, multi-centre study conducted between June 2018 and July 2019 in veterinary clinics from Chişinău, and in public dog shelters from the south (Cahul, Moldova) and central (Chişinău, Moldova) parts of the Republic of Moldova. All the owned dogs had a mixed indoor–outdoor lifestyle, while stray dogs living in shelters had an outdoor-only lifestyle. Owned dogs occasionally received ectoparasite preventive treatment, whereas the stray dogs were free of any prophylactic treatment as confirmed by shelter personnel. Before study inclusion, informed consent forms were obtained from the owners of the dogs or from the managers of the shelters. 

From each dog, a blood sample was collected by cephalic venipuncture. A 5-mL syringe and a sterile 18 to 22-gauge needle size were used according to the dog size. Samples of 2 mL were transferred in labeled tubes with anticoagulant (ethylenediaminetetraacetic acid) and stored at −20 °C for subsequent testing of TBPs. Location, sex, age, breed, origin, and travel history were recorded for each dog to assess the risk factors for tick-borne pathogens.

This study (project number: PD35/2018) was approved by the Ethical Committee of the University of Agricultural Sciences and Veterinary Medicine Cluj-Napoca, Cluj, Romania.

### 2.2. Molecular Techniques for Vector-Borne Pathogen Identification

Genomic DNA was extracted from 200 μL of whole blood using a commercial kit (Isolate II Genomic DNA Kit, meridian Bioscience, London, UK) according to the manufacturer’s instructions. In order to assess and exclude possible contaminations, in each set of extractions, a blank consisting of PCR-grade water was included and subsequently evaluated for the presence of nucleic acids using a NanoDrop spectrophotometer. The DNA samples were screened for the presence of various vector-borne pathogens using the previously published primers and protocols (Table 1). Each amplification set included a positive control consisting of pathogen DNA attained and confirmed by sequencing during previous studies, one negative control (DNA from a healthy dog), and one no-template control (NTC) consisting of PCR-grade water. 

The PCR products and controls were visualized by electrophoresis in 2% agarose gels stained with the RedSafe™ 20,000× Nucleic Acid Staining Solution (iNtRON Biotechnology, Seongnam, Korea), and their molecular weight was assessed via comparison with a molecular marker (HyperLadder™ 100 bp, meridian Bioscience, London, UK). All bands of the expected size were excised from the gels and purified using a commercial kit (Isolate II PCR and Gel Kit, meridian Bioscience, London, UK). The purified products were sequenced using an external service (performed by Macrogen Europe B.V., Amsterdam, The Netherlands). The obtained sequences were compared to those available in the GenBank^®^ database by means of Basic Local Alignment Search Tool (BLAST) analysis.

### 2.3. Statistical Analysis

Data were analysed using the statistics software Epi Info™ version 3.5.1. (CDC, 2008) [27]. The demographic and clinical characteristics of the sampled dogs were analysed descriptively. The frequency and prevalence of TBPs were tabulated with 95% confidence intervals (CIs). A multilogistic regression model was used to determine whether age, sex, breed, location, and origin were risk factors for infection with TBPs. The level of significance was set at *p*-value ≤ 0.05. Odds ratios with 95% CIs were calculated to compare the magnitude of the risk factor.

## 3. Results

### 3.1. Study Population

Of 120 dogs included in this study, 30 originated from veterinary clinics in Chişinău and 90 from public dog shelters from Cahul (n = 42) and Chişinău (n = 48). 

The mean age (± SD) of the dogs was 3.6 ± 2.4 years. Other demographic characteristics are shown in Table 2. Vector-borne pathogens were evaluated in blood samples collected from all the dogs.

### 3.2. Molecular Analysis Outcomes

TBPs were identified by PCR in 41/120 (34%) dogs. The overall prevalence of pathogen species identified in blood samples is shown in Figure 1. All blood samples tested negative for the following pathogens: *Borrelia* spp., *Rickettsia* spp., *F. tularensis*, *A. platys*, and *Ehrlichia canis*.

*Hepatozoon canis* was the most prevalent pathogen, detected in 19 dogs. Five unique DNA sequences of *H. canis* were obtained. Of these, three were detected from samples collected from three different dogs, the fourth was identified in three dogs, and the fifth in the remaining thirteen dogs. Three of the DNA sequences showed 100% identity similarity with other *H. canis* isolates from Europe (e.g., KX 712129, MN791088, KY693670, and KU893127), while the other two showed a similarity of 99.25% and 99.81%, respectively.

*B. canis* was detected in 14 blood samples, and four unique DNA sequences were obtained. Of the latter, one was present in a single dog and was 100% identical to a DNA sequence isolated from bat blood in Romania (MK934420). The remaining sequences were 99.6% (one dog), 99.81% (one dog), and 100% (11 dogs) identical to *B. canis* isolates from *Dermacentor reticulatus* ticks and dogs from Romania (HQ662634, MK836022). 

*A. phagocytophilum* was detected in seven positive dogs. Five unique sequences were obtained (one of them being present in three dogs) with a 99.8–100% nucleotide identity to *A. phagocytophilum* isolates originating from ticks and hosts throughout Europe (e.g., MF372791, KF312357, MW272752, and MW013537).

The *Bartonella* sp. sequence showed the highest similarity (99.7%) with two unnamed *Bartonella* sp. isolated from dogs in China (FJ464163, DQ192515), followed by *B. vinsonii* subsp. *Berkhoffii* isolated from dog blood in Sweden (99.41%; CP003124). 

All sequences obtained in this study were deposited in the GenBank^®^ database, under the following Accession Numbers: OP412819-OP412823 (*H. canis*), OP412824-OP412827 (*B. canis*), OP428642-OP428646 (*A. phagocytophilum*), and OP428647 (*Bartonella* sp.).

### 3.3. Risk Factors for Tick-Borne Pathogens 

Analysis of the risk factors revealed that the origin of dogs had a significant influence on A. phagocytophilum infection only; dogs originating from the veterinary clinic had a higher prevalence of this TBD than those from the shelters (16.6% versus 2.2%, respectively, *p* = 0.0292; OR: 27.0 [1.4–521.9]). Location had a significant influence only on H. canis infection: Dogs from Chișinău had a higher prevalence of H. canis infection as compared to those from Cahul (19.2% versus 9.5%, respectively, *p* = 0.0295; OR: 3.9 [95% CI: 1.1–13.4]). No significant differences were observed with regard to age, sex, and breed, regardless of the pathogen (Table 3).

## 4. Discussion

The Republic of Moldova is one of the European countries with major gaps in TBD surveillance, due to limited access to diagnostic tools, misdiagnosis, and low awareness with regard to some diseases. This study is the first to investigate the prevalence of *A. phagocytophilum*, *A. platys*, *B. burgdorferi*, *B. canis*, *Bartonella* spp., *E. canis, F. tularensis*, *H. canis*, and *Rickettsia* spp. in dogs from this country.

*H. canis* was the most prevalent TBP (15.8% [19/120]) identified in our study, which is an expected finding given that this protozoan parasite is widely spread in Europe and is endemic in the Mediterranean region and southeastern countries [28,29,30]. The infection rate recorded in our study is similar to those reported in south-central Romania (15% [14/96]) and Croatia (12% [108/924]), but much lower than in southern Romania (47% [163/300]) [28,31,32]. In Hungary, the prevalence is reported as higher (26% [33/126]), with different infection rates observed for shepherd (31%), hunting (8%), and stray (7%) dogs [33]. In Italy, the reported prevalence of *H. canis* varies from 3.6% (14/385) to 32.5% (38/117), depending on the region, the origin of dogs (kennel or hunting), and period of collection [29,34,35]. Similarly, in our study, the significant difference between the infection rate of the dogs from Chișinău (19.2 [15/78]) and those from Cahul (9.5% [4/42]) (*p* = 0.0295; OR: 3.9) could be explained by the different sampling periods and the spread of *R. sanguineus* s.l., the main vector of this pathogen [36]. Its presence in canids has been previously reported in the Chișinău area [37]. Other tick species are also thought to be competent vectors for *H. canis* [38,39]. The indoor biology of *R. sanguineus* s.l. may explain why *H. canis* was recorded in all but one dog from public shelters [40].

*H. canis* infection is usually asymptomatic in dogs. Clinical signs and symptoms, such as fever, anorexia, weight loss, lymphadenomegaly, anaemia, and lethargy, may be present when a high level of parasitemia is reached or if co-infections with other vector-borne pathogens occur [41]. Co-infections are highly likely to be detected given that the main vector of *H. canis*, *Rhipicephalus sanguineus*, can transmit various individual pathogens [41,42]. 

For *B. canis* infection, we recorded a prevalence of 11.7% (14/120), which is comparable with the prevalence recorded in a recent comprehensive survey of stray dogs from the south of Romania (9.6% [29/300]) and in a study of suburban and rural dogs in Serbia (13.5% [15/111]) [32,43], and lower than in non-symptomatic dogs screened for tick-borne infections in central Poland (25.3% [21/82]) [44]. The *B. canis* infection rate was substantially higher when symptomatic dogs were screened: 71.4% (174/49) in Romania and 87.8% (108/123) in Lithuania [45]. The differences in prevalence across European countries are related to the presence of *Dermacentor reticulatus* tick species, its geographical distribution, and its abundance in certain regions [46,47]. 

The importance of *B. canis* infection resides in the moderate to severe disease caused in dogs, which usually manifests with anaemia and haemolysis, fever, vomiting, lymphadenomegaly, hypotension, nephropathy, weight loss, and lethargy, and may progress to multi-organ failure with a high risk of mortality if diagnosis and specific treatment are delayed [46,48]. Máthé and colleagues reported a mortality rate of approximately 50% in dogs with acute kidney injury as a complication of *B. canis* [49], which highlights the importance of the specific treatment being started as early as possible in the course of the disease.

We found an *A. phagocytophilum* infection rate of 5.8% (7/120), which is comparable with the prevalence reported in Romania (5.3% [19/357] to 6.2% [16/257]), Slovakia (8% [6/87]), and Italy (up to 6% [3/50]), and lower than that recorded in southern Hungary (11% [14/100]) and Poland (up to 14% [13/92]) [44,50,51,52,53,54,55,56,57,58]. Lower infection rates have been reported in Tirana (1.0% [6/602]), Belgrade municipalities (0.0% [0/111]), and in the Czech Republic (3.4% [10/296]) [43,59,60]. Interestingly, in our study, the infection rate was significantly higher in owned dogs (16.6% [5/30]) compared to those originating from shelters (2.2% [2/90]). This could be explained by the limited access of shelter dogs to natural environments habited by *I. ricinus*.

The first report of *A. phagocytophilum* in the Republic of Moldova dates back to 2013, when this pathogen was identified at a prevalence rate of 2.4% (3/126) in the museum-archived *I. ricinus* female ticks collected in Moldova in 1960 [61]. Since then, *A. phagocytophilum* was identified at a prevalence ranging from 6.9% (18/262) to 19% (19/131) in ticks collected from migratory birds [14,62]. While these data highlight that *A. phagocytophilum* is an emerging pathogen in the Republic of Moldova, our study demonstrated that its transmission to dogs already occurred, and that the pathogen circulates among ixodid ticks from highly populated areas of the country. Given that *A. phagocytophilum* is the causative agent of canine granulocytic anaplasmosis in domestic carnivores, tick-borne fever in ruminants, equine granulocytic anaplasmosis in horses, and human granulocytic anaplasmosis [63], our study highlights the need for raising awareness of this potential disease amongst veterinarians and public health personnel. 

*Bartonella* spp. infection was found in one dog originating from a shelter in Cahul. The overall prevalence of *Bartonella* spp. of 0.8% was similar to that recorded in a study in Poland (0.3% [1/242]) [58], but lower than that reported in Greece (4% [2/50]), Italy (12% [7/60]), and Spain (27% [8/36]) [64,65]. In a recent study conducted in Portugal, all blood samples collected from dogs (25) tested negative for *Bartonella* spp. [66]. Even though the majority of acute *Bartonella* infections are self-limiting, in some cases, persistent infections can lead to various pathologic conditions in dogs and humans. Dogs can be accidental hosts for *B. vinsonii subsp. berkhoffii, B. henselae,* and other *B. species* [67]. *Bartonella* infection is more likely to manifest in dogs compared to cats, with clinical signs and symptoms such as fever, endocarditis and myocarditis, cardiac arrhythmias, pyogranulomatous lymphadenitis, hepatitis and pulmonary nodules, dermatitis, panniculitis, granulomatous rhinitis, and epistaxis [67,68]. It remains unknown whether dogs can transmit infection to humans [67]. Further studies are needed to fill this gap, especially since *Bartonella* spp. pose a zoonotic risk for immunosuppressed as well as immunocompetent individuals [68].

All samples tested negative for *A. platys* and *E. canis*. This was expected given that the transmission route of both pathogens is restricted to Mediterranean countries [40,69,70]. *R. sanguineus* ticks should be collected and tick DNA analysed to better understand the *E. canis* distribution in the Republic of Moldova and the vectorial role of *R. sanguineus* s.l.’s temperate lineage in the maintenance and transmission of this pathogen [71]. However, potential infection with *A. platys* should be kept in mind for dogs presenting with cyclic thrombocytopenia [72], especially if increased travel activities are reported by owners or if dogs were imported from endemic areas. Similarly, *E. canis*-related TBD should be kept on the radar due to its clinical importance in dogs. *E. canis* causes canine monocytic ehrlichiosis, which manifests with clinical (e.g., fever, anorexia, diarrhoea, vomiting, lymphadenopathy, petechial haemorrhages, bleeding tendency) and haematological abnormalities (e.g., anaemia and thrombocytopenia), or can determine chronic infection [69]. 

Our study’s main strength is its novelty; this is the first study to report the presence of *A. phagocytophilum*, *B. canis*, and *Bartonella* spp. (assessed by PCR from blood samples) in dogs from the Republic of Moldova. Even if the set of inclusion and exclusion criteria was minimal, the sample size of our study was limited. This relatively small sample size did not allow us to draw definitive conclusions or identify fair risk factors. Another limitation of our study originates in the screening method used for TBP identification, i.e., the detection of the pathogens’ genomic DNA in blood by PCR occurs more easily in the acute phase of a disease. However, previous exposure to any pathogen that may have been present but was cleared by the time of blood sampling was not assessed (e.g., serological tests). 

## 5. Conclusions

Our study reported, for the first time, the presence of *A. phagocytophilum*, *H. canis*, *B. canis*, and *Bartonella* spp. in dogs from the Republic of Moldova, extending the knowledge about the geographic distribution of these TBDs. Suitable actions should be taken to prevent or limit the spread of TBDs in this country and to raise awareness among veterinarians and public health personnel. As a preventive action for the limitation of TBD spread, we would recommend routine use of external deworming and acaricides in both owned dogs and those from public shelters. Further surveillance studies on TBDs in dog populations are needed.

## Figures and Tables

**Figure 1 pathogens-11-01267-f001:**
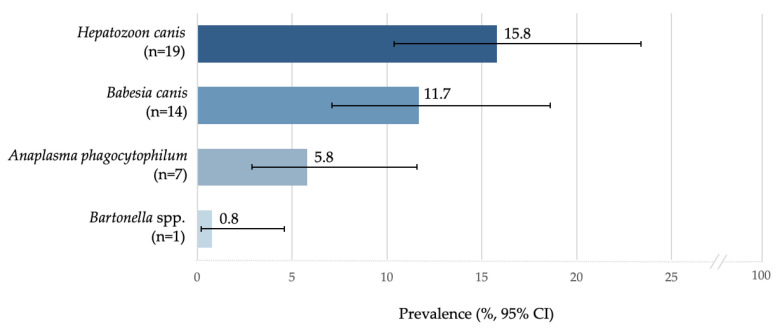
Overall prevalence of tick-borne pathogens in dogs from the Republic of Moldova. n, total number of positive samples for a specific pathogen; CI, confidence interval.

**Table 1 pathogens-11-01267-t001:** List of vector-borne pathogens tested by PCR and the protocol used.

Pathogen	Target Gene	Product Size (bp)	Forward Primer	Reverse Primer	References
Piroplasmids and *Hepatozoon* spp. (Nested PCR)	18S rRNA	561–620	BTH-1 F: CCT GAG AAA CGG CTA CCA CAT CT	BTH-1R: TTG CGA CCA TAC TCC CCC CA	[18]
			GF2: GTC TTG TAA TTG GAA TGA TGG	GR2: CCA AAG ACT TTG ATT TCT CTC	
*Borrelia* spp. (Nested PCR)	*flaB*	350	FlaLL: ACA TAT TCA GAT GCA GAC AGA GGT	FlaRL: TGT TAG ACG TTA CCG ATA CTA ACG	[19,20]
			FlaLS: AAC AGC TGA AGA GCT TGG AAT G	FlaRS: CGA TAA TCT TAC TAT TCA CTA GTT TC	
SFG *Rickettsia*	*gltA*	381	Rsfg877: GGG GGC CTG CTC ACG GCG G	Rsfg1258: ATT GCA AAA AGT ACA GTG AAC A	[21]
*Bartonella* spp.	*gltA*	380–400	bart781: GGG GAC CAG CTC ATG GTG G	bart1137: AAT GCA AAA AGA ACA GTA AAC A	[22]
*Francisella tularensis*	17-kDa lipoprotein gene	400	TUL4-435: GCT GTA TCA TCA TTT AAT AAA CTG CTG	TUL4-863: TTG GGA AGC TTG TAT CAT GGC ACT	[23]
*Anaplasma phagocytophilum* (Heminested PCR)	*groEL*	570	EphplgroEL(569)F: ATG GTA TGC AGT TTG ATC GC	EphplgroEL(1193)R: TCT ACT CTG TCT TTG CGT TC	[24]
				EphgroEL(1142)R: TTG AGT ACA GCA ACA CCA CCG GAA	
*Anaplasma platys*	16S rRNA	349	EPLAT5: TTT GTC GTA GCT TGC TAT GAT	EPLAT3: CTT CTG TGG GTA CCG TC	[25]
*Ehrlichia canis* (Nested PCR)	16S rRNA	389	ECC: AGA ACG AAC GCT GGC GGC AAG CC	ECB: CGT ATT ACC GCG GCT GCT GGC A	[26]
			«canis»: CAA TTA TTT ATA GCC TCT GGC TAT AGG A	HE3: TAT AGG TAC CGT CAT TAT CTT CCC TAT	

PCR, polymerase chain reaction; rRNA, ribosomal ribonucleic acid.

**Table 2 pathogens-11-01267-t002:** Demographic characteristics of the dogs included in the study.

Category	Sampled Dogs (N = 120) n (%)
Age (years)	
0 to 1	11 (9.2)
1 to 8	99 (82.5)
>8	10 (8.3)
Sex	
Female	70 (58.3)
Male	50 (41.7)
Breed	
Pure breed	11 (9.2)
Mixed	109 (90.8)
Location	
Chișinău	78 (65.0)
Cahul	42 (35.0)
Origin	
Shelter	90 (75)
Clinic	30 (25)

N, total number of dogs; n, number of dogs in a given category.

**Table 3 pathogens-11-01267-t003:** Prevalence (%) of tick-borne pathogens in dogs from the Republic of Moldova and analysis of the risk factors.

Category	*Hepatozoon canis*	*Babesia canis*	*Anaplasma* *phagocytophilum*	*Bartonella* spp.
Age, n (%)				
0 to 1, N = 11	2 (18.2)	2 (18.2)	2 (18.2)	0 (0.0)
1 to 8, N = 99	17 (17.2)	11 (11.1)	4 (4.0)	1 (1.0)
>8, N = 10	0 (0.0)	1 (10.0)	1 (10.0)	0 (0.0)
*p*-value	0.26	0.5	0.8	-
OR (95% CI)	0.5 (0.1–1.7)	0.6 (0.1–2.7)	0.8 (0.1–5.1)	-
Sex; n (%)				
Female, N = 70	11 (15.7)	6 (8.6)	4 (5.7)	0 (0.0)
Male, N = 50	8 (16.0)	8 (16.0)	3 (6.0)	1 (2.0)
*p*-value	0.7	0.2	0.6	-
OR (95% CI)	0.8 (0.3–2.4)	2.1 (0.6–6.7)	1.6 (0.2–10.3)	-
Breed, n (%)				
Pure breed, N = 11	0 (0.0)	0 (0.0)	3 (27.3)	0 (0.0)
Mixed, N = 109	19 (17.4)	14 (12.8)	4 (3.7)	1 (0.9)
*p*-value	1.0	1.0	0.1	-
OR (95% CI)	0.0 (0.0–>1.0 × 10 ^12^)	0.0 (0.0–>1.0 × 10^12^)	6.4 (0.5–74.8)	-
Location, n (%)				
Chișinău, N = 78	15 (19.2)	9 (11.5)	4 (5.1)	0 (0.0)
Cahul, N = 42	4 (9.5)	5 (11.9)	3 (7.1)	1 (2.4)
*p*-value	0.03	0.8	0.06	-
OR (95% CI)	3.9 (1.1–13.4)	1.2 (0.4–4.2)	0.04 (0.002–1.2)	-
Origin, n (%)				
Shelter, N = 90	18 (20.0)	13 (14.4)	2 (2.2)	1 (1.1)
Vet Clinic, N = 30	1 (3.3)	1 (3.3)	5 (16.7)	0 (0.0)
*p*-value	0.1	0.3	0.03	-
OR (95% CI)	0.2 (0.02–1.4)	0.3 (0.04–3.0)	27.0 (1.4–521.9)	-

N, total number of dogs in a given category; n, number of positive samples in a given category; OR, odd ratio; CI, confidence interval; -, not applicable as only one sample tested positive. Bolded values indicate statistical significance.

## Data Availability

The data presented in this study are available on request from the corresponding author.

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
