# Peer review of "Epidemiological Survey of the Main Tick-Borne Pathogens Infecting Dogs from the Republic of Moldova"

_pathogens, 2022, doi:10.3390/pathogens11111267_

Round 1

Reviewer 1 Report

This paper identifies the tick-borne pathogens and associated risk factors in dogs from Moldova. It is relevant since it is the first report of these pathogens in mammalian hosts and it supports the findings of that done in ticks in previous studies. The authors have identified some gaps in the research and attempted to answer a few. Overall the paper was well written and comprehensive. I have attached a few comments.

Reviewer 2 Report

The manuscript concerning an "Epidemiological survey of the main tick-borne pathogens infecting dogs from the Republic of Moldova" is well-written, methodology used is adequate and results are well-presented and discussed.

No major suggestion were identified.

Minor suggestions:

-the text must be formatted and some words suchs as "versus" or "vs" must be in italic

- in the abstract and introduction the full name of parasite should be given, such as Babesia canis.

- the number of references appears to be too high - please consider reviewing this list

- despite of the above mentioned the inclusion of references such as Sanches GS (2018) and Torrejon E (2022) would improve discussion

- authors mention the work was conducted under a multi-center study but an explanation of why are researchers from Romania handling samples collected in the Republic of Moldova. Where the samples collected by the authors? Or where they obtained by people living in Moldava? 

- investigators and institutions of the republic of Moldova are acknowledged; please introduce which were their contributions

Reviewer 3 Report

Please see attachment. It's not clear if the author's forgot to report controls (minor revision) or to include controls (major revision).
